# The Proinflammatory Role of ANGPTL8 R59W Variant in Modulating Inflammation through NF-κB Signaling Pathway under TNFα Stimulation

**DOI:** 10.3390/cells12212563

**Published:** 2023-11-02

**Authors:** Mohamed Abu-Farha, Dhanya Madhu, Prashantha Hebbar, Anwar Mohammad, Arshad Channanath, Sina Kavalakatt, Nada Alam-Eldin, Fatima Alterki, Ibrahim Taher, Osama Alsmadi, Mohammad Shehab, Hossein Arefanian, Rasheed Ahmad, Thangavel Alphonse Thanaraj, Fahd Al-Mulla, Jehad Abubaker

**Affiliations:** 1Department of Biochemistry and Molecular Biology, Dasman Diabetes Institute, Dasman 15462, Kuwait; mohamed.abufarha@dasmaninstitute.org (M.A.-F.); dhanya.madhu@dasmaninstitute.org (D.M.); anwar.mohammad@dasmaninstitute.org (A.M.); sina.kavalakatt@dasmaninstitute.org (S.K.); nada.alamaldin@dasmaninstitute.org (N.A.-E.); 2Department of Genetics and Bioinformatics, Dasman Diabetes Institute, Dasman 15462, Kuwait; prashantha.hebbar@dasmaninstitute.org (P.H.); arshad.channanath@dasmaninstitute.org (A.C.); fahd.almulla@dasmaninstitute.org (F.A.-M.); 3Department of internal Medicine, Amiri Hospital, Ministry of Health, Kuwait City 15462, Kuwait; fatima.alterki@gmail.com; 4Microbiology Unit, Department of Pathology, College of Medicine, Jouf University, Sakaka P.O. Box 2014, Saudi Arabia; itaher@ju.edu.sa; 5Department of Cell Therapy and Applied Genomics, King Hussein Cancer Center, Amman 1269, Jordan; oa.12163@khcc.jo; 6Division of Gastroenterology, Department of Internal Medicine, Mubarak Alkabeer University Hospital, Kuwait University, Kuwait City 47061, Kuwait; dr_mshehab@hotmail.com; 7Department of Immunology & Microbiology, Dasman Diabetes Institute, Dasman 15462, Kuwait; hossein.arefanian@dasmaninstitute.org (H.A.); rasheed.ahmad@dasmaninstitute.org (R.A.)

**Keywords:** ANGPTL8, TNFα, NF-κB, glucose metabolism, single nucleotide polymorphism

## Abstract

Background: Angiopoietin-like protein 8 (ANGPTL8) is known to regulate lipid metabolism and inflammation. It interacts with ANGPTL3 and ANGPTL4 to regulate lipoprotein lipase (LPL) activity and with IKK to modulate NF-κB activity. Further, a single nucleotide polymorphism (SNP) leading to the ANGPTL8 R59W variant associates with reduced low-density lipoprotein/high-density lipoprotein (LDL/HDL) and increased fasting blood glucose (FBG) in Hispanic and Arab individuals, respectively. In this study, we investigate the impact of the R59W variant on the inflammatory activity of ANGPTL8. Methods: The ANGPTL8 R59W variant was genotyped in a discovery cohort of 867 Arab individuals from Kuwait. Plasma levels of ANGPTL8 and inflammatory markers were measured and tested for associations with the genotype; the associations were tested for replication in an independent cohort of 278 Arab individuals. Impact of the ANGPTL8 R59W variant on NF-κB activity was examined using approaches including overexpression, luciferase assay, and structural modeling of binding dynamics. Results: The ANGPTL8 R59W variant was associated with increased circulatory levels of tumor necrosis factor alpha (TNFα) and interleukin 7 (IL7). Our in vitro studies using HepG2 cells revealed an increased phosphorylation of key inflammatory proteins of the NF-κB pathway in individuals with the R59W variant as compared to those with the wild type, and TNFα stimulation further elevated it. This finding was substantiated by increased luciferase activity of NF-κB p65 with the R59W variant. Modeled structural and binding variation due to R59W change in ANGPTL8 agreed with the observed increase in NF-κB activity. Conclusion: ANGPTL8 R59W is associated with increased circulatory TNFα, IL7, and NF-κB p65 activity. Weak transient binding of the ANGPTL8 R59W variant explains its regulatory role on the NF-κB pathway and inflammation.

## 1. Introduction

Obesity and type 2 diabetes (T2D) have become serious global health conditions that have reached alarming levels [1]. They are associated with impaired metabolism leading to high circulating free fatty acids (FFAs) and triglycerides (TG) [2]. Another outcome of T2D is the inability of the pancreas to meet the increased insulin demand leading to high glucose concentration in the blood circulation [3]. A family of proteins that is structurally similar to angiopoietins has been identified as angiopoietin-like proteins (ANGPTLs) and has been demonstrated to play an important role in gluco-lipid homoeostasis. They are a group of secreted glycoproteins composed of eight members (ANGPTL1-8). All of them contain an amino-terminal, a linker region, and a carboxy-terminal fibrinogen-like domain. However, ANGPTL8 lacks a fibrinogen/angiopoietin-like domain [4]. ANGPTL8, also known as lipasin, RIFL, TD26, or betatrophin, is a 22 KDa protein comprising 198 amino acids and is a distinct member of the angiopoietin-like proteins family. Many of the ANGPTLs are involved in multiple biological processes such as lipid metabolism, inflammation [5], and hematopoietic stem cell activity [6,7].

ANGPTL8 is abundant in liver and fat tissues, particularly in brown adipose tissue (BAT) and white adipose tissue (WAT) [8]. Nutritional regulation has been advocated as an important component of ANGPTL8 expression [9]. During fasting, ANGPTL8 expression was reduced by 80% in BAT and WAT, whereas mice exposed to a high fat diet significantly increased ANGPTL8 mRNA expression in the liver and BAT [8]. In human obese subjects with metabolic syndrome, plasma ANGPTL8 levels have been inversely correlated with protein intake [10]. Hence, ANGPTL8 levels are greatly modulated by the amount of protein intake. During fasting conditions, glucocorticoids are increased, and they suppress ANGPTL8 expression through activation of a glucocorticoid receptor that binds to the negative glucocorticoid response element (nGRE) [11].

Several studies have reported that ANGPTL8 acts as a key modulator in lipid metabolism and metabolic disorders [12,13]. ANGPTL8 has at least two physiological roles, namely those of regulating plasma TG levels and controlling inflammation. Overexpression of ANGPTL8 in mice increased TG levels more than fivefold [8], whereas its deficiency reduced TG by twofold [14]. ANGPTL8 deficiency increases postprandial lipoprotein lipase (LPL), especially in cardiac and skeletal muscles, and a proposed mechanism was shown to demonstrate ANGPTL8 on TG metabolism in different nutritional states wherein ANGPTL3, ANGPTL4, and ANGPTL8 coordinate to regulate TG trafficking [15]. Research has shown that circulating levels of ANGPTL8 were elevated in arteriosclerosis and non-alcoholic steatosis [16,17]. Plasma levels of ANGPTL8 were elevated in patients with severe infections. Strong correlations between circulating ANGPTL8 and LPS-induced acute inflammatory response have been demonstrated in animal models [18]. Henceforth, the impact of ANGPTL8 on the regulation of blood lipoproteins throws light on its role in modulating inflammation.

Nuclear factor kappa-light-chain-enhancer of activated B cells (NF-κB) is a key transcription factor that is implicated in inflammatory signaling cascade. It plays a significant role in many physiological and pathological processes including inflammation, immunity, and metabolism [19]. Under non-stimulated conditions, NF-κB is inactive in the cytosol as it binds to inhibitors of IκBα. When stimulated using TNFα or any proinflammatory cytokine, it causes phosphorylation and degradation of IκBα. This releases active NF-κB and causes it to translocate to nuclei followed by the induction of the expression of downstream targets [20]. TNFα is the most classical model to study the NF-κB pathway. The TNFα-induced NF-κB activation must be tightly controlled to avoid inflammatory diseases, autoimmunity, and cancers [21,22]. Zhang et al. have shown that ANGPTL8 acts as a co-receptor of p62 in selective autophagy and that the interaction between p62 and ANGPTL8 are mutually required for IKKγ degradation [18]. Another group suggested that ANGPTL8 promoted ECM degradation and inflammatory cytokine release through activating the NF-κB signaling pathway which displays the detrimental role of ANGPTL8 in intervertebral disc degeneration (IDD). They demonstrated an enhanced NF-κB signaling pathway in TNF-α treated Nucleus pulposus (NP) cells through increased expression of ANGPTL8 resulting in ECM degradation and inflammation whereas inhibition of ANGPTL8 resulted in attenuated activation of NF-κB signaling [23]. Several genome-wide association studies, as listed in the GWAS Catalog (https://www.ebi.ac.uk/gwas/ accessed on 20 February 2023), have associated the ANGPTL8 rs2278426_ c.175C.T_p.R59W variant with the lipid traits of total cholesterol, low-density cholesterol, high-density cholesterol, and triglycerides in global populations such as those of European ancestry, East Asian ancestry, and Hispanic ancestry, as well as in multi-ethnic cohorts which include various additional ethnicities such as African American and South Asian. The variant is also listed as associated with waist-hip ratios in individuals of European ancestry. Although the role of ANGPTL8 was established in the context of inflammation, the impact of the R59W ANGPTL8 variant in regulating inflammation in vitro has not been explored so far to our knowledge. Therefore, our objective was to investigate the differential effect of the ANGPTL8 R59W variant over the wild type (WT) in modulating inflammatory pathways.

In this study, we aimed to assess the association of the R59W variant with various clinical lipid and inflammatory traits in a discovery cohort of 867 Arab individuals and a replication cohort of 278 Arab individuals in order to decipher the functional consequences of these association signals using a hepatocyte cell line model under non-stimulated and stimulated TNFα treatment. The impact of overexpressing the ANGPTL8 variant on NF-ĸB pathway activity and downstream markers was assessed. Further, structural and binding dynamics of the ANGPTL8 R59W variant and the wild type were investigated.

## 2. Materials and Methods

### 2.1. Recruitment of Participants and Study Cohort

The study protocol was reviewed and approved by the Ethical Review Committee of the Dasman Diabetes Institute as per the guidelines of the Declaration of Helsinki and of the US Federal Policy for the Protection of Human Subjects. A discovery cohort of 867 participants and an independent replication cohort of 278 participants were recruited. The discovery cohort comprised adults (>18 years of age) of Arab ethnicity across the six governorates of the State of Kuwait recruited by way of using a stratified random sampling of people from the computerized register of Kuwait’s Public Authority of Civil Information (PACI). The replication cohort was created by way of recruiting from the general public that visits our institute for facilities such as the physical fitness center and tertiary medical care clinics.

Briefly, native adult Kuwaiti individuals of Arab ethnicity were recruited as study subjects. Pregnant women were excluded. Data on age, sex, health disorders (e.g., diabetes and hypertension), and baseline characteristics such as height, weight, waist circumference, and blood pressure were recorded for each participant upon recruitment. Information on whether the participant undergoes medication for lowering lipid levels or for treating diabetes and hypertension was recorded and was subsequently used to adjust the models for genotype-trait association tests. Every participant signed the informed consent form before participating in this study.

### 2.2. Blood Sample Collection and Processing

After confirming that the participant had fasted overnight, blood samples were collected in EDTA-treated tubes. DNA was extracted using the Gentra Puregene^®^ kit (Qiagen, Valencia, CA, USA) and was quantified using the Quant-iT™ PicoGreen^®^ dsDNA Assay Kit (Life Technologies, Grand Island, NY, USA) and the Epoch Microplate Spectrophotometer (BioTek Instruments, Santa Clara, CA, USA). Absorbance values at 260–280 nm were checked for adherence to an optical density range of 1.8–2.1.

### 2.3. Estimation of Plasma Levels of Various Biomarkers

Plasma was separated from blood samples by centrifugation and was aliquoted and stored at −80 °C. ANGPTL8 was measured as previously reported [24,25] using the ELISA kit from EIAab Sciences (Wuhan, China, Cat# E1164H). Briefly, the samples were diluted ten times using the sample diluent provided in the kit. The standards (recombinant protein of known concentration) provided in the kit was reconstituted and diluted as per kit protocol. Diluted samples and standards were loaded onto the plate and the procedure was followed as per the instructions provided in the kit. The absorbance was read at 450 nm on the Synergy H1 plate reader (BioTek, Vermont, VT, USA). The concentrations of the unknown samples were determined by Gen 5 (v 5.1) based on a 5 PTL standard curve. TNFα, interleukins, and ghrelin were measured using Bioplex multiplex kits (cat# M50-0KCAFOY and cat # 171-A4S01M, Bio-Rad, Hercules, CA, USA). Standards with known concentrations of specific analytes were reconstituted and diluted as instructed in the manual. The assay was performed, and the plasma levels of the various analytes were determined using the Bio-plex 200 system (Bio-Rad, Hercules, CA, USA). The concentrations of the unknown samples were calculated by the Bio-plex Manager software (v 6.0) based on a 5 PTL standard curve.

### 2.4. Targeted Genotyping of the ANGPTL8 Study Variant R59W

Candidate SNP genotyping was performed using the TaqMan^®^ Genotyping Assay kit on the ABI 7500 Real-Time PCR System from Applied Biosystems (Foster City, CA, USA). Each polymerase chain reaction sample was composed of 10 ng of DNA, 5× FIREPol^®^ Master Mix (Solis BioDyne, Tartu, Estonia), and 1 µL of 20× TaqMan^®^ SNP Genotyping Assay. Thermal cycling conditions were set at 60 °C for 1 min and at 95 °C for 15 min followed by 40 cycles of 95 °C for 15 s and 60 °C for 1 min. Sanger sequencing was performed using the BigDye™ Terminator v3.1 Cycle Sequencing on an Applied Biosystems 3730xl DNA Analyzer for selected cases of homozygotes and heterozygotes to validate the genotypes determined by the candidate genotyping assay.

### 2.5. Quality Check Procedures for SNP and Trait Measurements

We used PLINK (version1.9) [26] to assess the SNP quality and the trait variance. We calculated minor allele frequency (MAF) and Hardy–Weinberg equilibrium for the study variant. Any quantitative trait value < Q1 − 1.5 × IQR or any value > Q3 + 1.5 × IQR was considered as an outlier and was excluded from further statistical analysis.

### 2.6. Allele-Based Association Tests and Thresholds for Ascertaining Statistical Significance

Allele-based statistical association tests for the study variant with the quantitative traits and biomarker levels were performed using linear regression analyses adjusting for regular corrections toward age and sex. We also adjusted for diabetes medication and lipid-lowering medication. A threshold of <0.05 was set for the *p*-value to assess the statistical significances.

### 2.7. Cell culture, Transfection, and Treatment

HepG2 cells were procured from the American Type Culture Collection (Rockville, Baltimore, MD, USA) and cultured in Minimum Essential Medium (MEM) supplemented with 10% fetal bovine serum (FBS) and penicillin/streptomycin. Cells were seeded to an 80% confluency and proceeded for transfection. Wild type and R59W clones (Blue Heron Biotech, OriGene, Rockville, MD, USA) of ANGPTL8 with Myc-DDK tags were used for transfection using Lipofectamine 3000 (Invitrogen, Carlsbad, CA, USA) with 5 µg of plasmid for 48 h. Myc tagged pCMV6 vector (OriGene, Rockville, MD, USA) was used as a control for the transfection experiments. After 48 h following transfection, the cells were treated with Recombinant Human TNFα Protein (R&D Systems, Minneapolis, MN, USA) at 25 ng/mL for 18 h. The cells were then processed for protein analysis.

### 2.8. Western Blot Analysis

Protein extraction was conducted using RIPA buffer (50 mM Tris-HCl, pH 7.5, 150 mM NaCl, 1% Triton X-100, 1 mM EDTA, 0.5% sodium deoxycholate, and 0.1% SDS) for the cell extracts of transfected cell lysates with/without human TNFα stimulation. The Bradford method was used to estimate protein concentrations normalized to β-globulin. In total, 20µg of protein samples were prepared in loading buffer containing β-mercaptoethanol and resolved on 10% SDS-PAGE gels. Then, proteins were transferred onto PVDF membranes (100 V for 75 min) and blocked for 2 h at room temperature (RT) using 5% non-fat dried milk in Tris-buffered saline containing 0.05% Tween 20. Membranes were then probed with the primary antibodies P- NF-κB-P65 (3033, Cell signaling), Total NF-κB (8242s, Cell signaling), Phospho-IKKα/β (Ser176/180) (2697, Cell signaling), Total IKKα (2682, Cell signaling), Phospho-IκBα (Ser32) (2859, Cell signaling), Total IκBα (9242, Cell signaling), IL6 (ab6672, abcam), TNFα (ab6671,abcam) at 4 °C for overnight incubation. The membranes were then washed followed by incubation with Rabbit Horseradish Peroxidase (HRP) conjugated secondary antibody (1:10,000 dilution) for 2 h at room temperature. Protein bands were visualized by chemiluminescence using super sensitivity West Femto ECL reagent (Thermo Scientific, Rockford, IL, USA), gel images were captured using the Versadoc 5000 system (Bio-Rad, Hercules, CA, USA), and band intensities were measured using Quantity One Software (Bio-Rad, Hercules, CA, USA). GAPDH was used as internal control for protein loading and detected using anti-GAPDH antibody (ABS16; Millipore, Temecula, CA, USA).

### 2.9. Luciferase Activity

HepG2 cells were cultured in Minimum Essential Medium (MEM) supplemented with 10% fetal bovine serum (FBS) and penicillin/streptomycin and seeded in twenty-four well plates. Reporter plasmid carrying luciferase gene under the control of NF-κB response elements was co-transfected with the expression vector encoding ANGPTL8 (pCMV-ANGPTL8), its mutant (ANGPTL8-R59W), and the pCMV empty vector (OriGene Technologies, Inc, Rockville, MD, USA). Briefly, the 3xwt-κB-pGL3 plasmid was constructed by inserting three copies of the wild type (5′-AGTTGAGGGGACTTTCCCAGGCTG-3′) NF-κB binding site into the unique Nhe-I and Xho-I restriction sites upstream of the SV40 minimal promoter of pGL3 vector (Promega, Madison, WI, USA). Luciferase assay (Promega Dual Luciferase assay, E1910) was performed 24 h post transfection as per the instructions in the kit. Briefly, the cells were lysed using the lysis buffer provided in the kit. The lysates were centrifuged for 30 s in a refrigerated microcentrifuge. Cleared lysates were used to perform the dual luciferase reporter assay protocol. The cell lysates were mixed with Luciferase reagent II (supplied in the kit) and were measured on a Synergy Hybrid H4 plate reader (BioTek, Winooski, VT, USA) and normalized according to protein concentration. Transfection efficiency was assessed using anti-FLAG antibodies as well as against the Renilla luciferase activity.

### 2.10. Structural Analysis of ANGPTL8 and Binding to IKKβ

The structure of ANGPTL8 was modeled with the SWISS-Model [27] and I-TASSER [28]. In this study, we utilized the X-ray crystal structure of the iSH2 domain of the Phosphatidylinositol 3-kinase (PI3K) p85β subunit (PDB: 3MTT), previously used by Siddiqa et al. [29]. To predict the effect of the R59W substitution on the ANGPTL8 structure, DynaMut web, a server that assesses the impact of variants on protein dynamics and stability (Rodrigues, Pires et al. 2021), was used. Furthermore, the modeled ANGPLT8 structure was minimized with YASARA [30], which finds the lowest energy conformation by reducing the steric energies between bond lengths and angles of both ANGPTL8 WT and ANGPTL8 R59W structures.

The effect of the ANGPTL8 variant on the binding to IKKβ was elucidated by interacting both the ANGPTL8 wild type (R59) and the mutant (W59) with IKKβ prior to modeling the ANGPTL8 protein-protein interactions with IKKβ, and the IKKβ interface binding residues for interacting with ANGPTL8 were projected with CPORT [31]. Protein-protein docking of the ANGPTL8 wild type (R59) and mutant (W59) with IKKβ PDB ID: 4KIK) was conducted using the HDOCK server [32], which is based on a hybrid algorithm of template-based modeling and ab initio free docking. Model 1, with the lowest docking energy score and the highest ligand RMSD, was selected to analyze binding energy scores (Kd) using the PRODIGY server [33]. Furthermore, PDBsum [34] protein-protein analysis was used to depict interface residues on the secondary structural elements of ANGPTL8 and IKKβ.

### 2.11. Power Calculation

We used the Quanto software tool (University of Southern California, Los Angeles, CA, USA) to calculate the power of the study cohorts and its ability to delineate quantitative trait variability at a given power (which was set at 80%). We considered additive mode as the underlying genetic model with “gene only” hypothesis at type 1 error, *p* ≤ 0.05. Genetic effect that accounts for at least 0.1–5% variance in the trait was detected by assuming RG 2 (estimate for marginal genetic effect) values in the range of 0.001–0.05 in step of 0.003. We considered the (mean ± standard deviation) of the quantitative trait and MAF from the study cohorts in these calculations.

## 3. Results

### 3.1. Study Cohorts

Two independent cohorts of Arab individuals from Kuwait were recruited for the discovery and replication phase. In total, 867 participants formed the discovery cohort and 278 participants formed the replication cohort.

### 3.2. Characteristics of the R59W Variant

The rs2278426 is a missense (R59W) variant in *ANGPTL8* (betatrophin), while it is an intronic variant in *DOCK6*. The variant downregulates a novel transcript CTC-510F12.4 in whole blood, upregulates *DOCK6* in Adipose-subcutaneous, and is reported to be associated with high-density lipoprotein (HDL), low-density lipoprotein (LDL), total cholesterol (TC), triglyceride (TG), and waist-hip ratio in the NHGRI-EBI GWAS Catalog [35]. In our study cohorts, the minor allele (T) occurred at a frequency of around 0.102 in the discovery cohort and 0.101 in the replication cohort. Distribution of individuals in terms of genotypes homozygous for reference allele (CC): heterozygous for reference and mutated allele (CT): homozygous for mutated allele (TT) was 701:155:11 in the discovery cohort and 227:46:5 in the replication cohort.

### 3.3. Characteristics of the Study Cohorts

Summary statistics on the clinical characteristics of the participants from the study cohorts are presented in Table 1A. The mean age of the participants was 43.36 ± 10.78 years in the discovery cohort and 46.25 ± 12.38 years in the replication cohort. The cohorts comprised mostly class I obese subjects with a mean body mass index (BMI) of 28.53 ± 4.9 (discovery cohort) and 29.93 ± 5.17 (replication cohort) kg/m^2^ and a mean waist circumference (WC) of 93.78 ± 11.51 (discovery cohort) and 99.36 ± 13.36 cm (replication cohort). Mean hemoglobin A1c (HbA1c) (5.47 ± 0.74 and 6.31 ± 1.3% in the discovery and replication cohorts, respectively), low-density lipoprotein (LDL) (3.34 ± 0.89 and 3.13 ± 0.96 mmol/L), high-density lipoprotein (HDL) (1.15 ± 0.29 and 1.2 ± 0.32 mmol/L), total cholesterol (TC) (5.19 ± 0.93 and 5.02 ± 1.09 mmol/L), and triglyceride (TG) (1.42 ± 0.61 and 1.22 ± 0.6 mmol/L) were normal or near optimal. Of the subjects from the discovery cohort, 38.2% were obese, 29.9% were diabetic, 22.7% were taking medication for diabetes, and 22.7% were taking medication for lowering lipids; these statistics differ considerably from the replication cohort wherein 48.7% were obese, 43.3% were diabetic, 36.6% were taking medication for diabetes, and 31.8% were taking medication for lowering lipids.

After partitioning the study cohorts based on the genotypes (CC versus CT + TT) at the rs2278426 variant, the discovery cohort exhibited significant differences (*p*-value ≤ 0.05) in the mean values for TNFα and IL7 as well as IL6 (Figure 1).

### 3.4. Association of the ANGPTL8 rs2278426 Variant with Increased Circulatory Levels of TNFα and IL7

The summary statistics of associations observed between the variant and the phenotype traits, derived from allele-based association tests based on additive models corrected for age and sex as well as for medications, are listed in Table 1B. The variant was associated with increased levels of tumor necrosis factor alpha (TNFα), interleukin 7 (IL7), interleukin 6 (IL6), and ghrelin in the discovery cohort; however, only the associations with TNFα and IL7 were seen replicating. The SNP passed the test for Hardy–Weinberg Equilibrium (HWE) in both the discovery and replication cohorts (Table 2) and thus the population meets the assumption of HW equilibrium.

In order to confirm that the impact of the effect allele at the variant on the levels of TNFα and IL7 is not due to obese or diabetic or hypertensive subjects in the cohort, we performed allele-based logistic regression analysis to evaluate the risk of disorders (diabetes, obesity, and hypertension) due to the variant (Table 3). Though the odds ratio (OR) values were notable for the disease status of diabetes and hypertension, the *p*-values were not significant.

### 3.5. R59W Variant Activates NF-κB Pathway Compared to the Wild Type

In order to assess the impact of the R59W variant on the activity of the NF-κB pathway, we conducted overexpression and luciferase binding assays using the HepG2 cell line. A reporter plasmid carrying a luciferase gene under the control of NF-κB was co-transfected with the expression vector encoding ANGPTL8 (pCMV-ANGPTL8) or its variant (ANGPTL8-R59W) (OriGene Technologies, Inc., Rockville, MD, USA). Overexpression of the variant resulted in significantly increased phosphorylation of both the IKKα/β protein and NF-κB p-65 when compared to the wild type supporting the activation of the NF-κB pathway, * *p* value < 0.05 (Figure 2B,C). Moreover, a significant increase (over twofold) in luciferase activity with the R59W variant compared to the wild type was observed (Figure 2A). These results indicate that the R59W variant of ANGPTL8 can activate the NF-κB pathway compared to the wild type. The expression of the flags for both the ANGPTL8 wild type and its variant following transfection are shown in Figure 2D.

### 3.6. R59W Variant further Elevates Activation of the NF-κB Signaling Pathway Compared to the Wild Type under TNFα Stimulation

Literature reports have shown that the transcription and expression of *ANGPTL8* were both significantly increased in HepG2 cells after being treated with TNFα in a dose-dependent fashion [18]. Therefore, we decided to assess the impact of ANGPTL8 on NF-ĸB under TNFα stimulation. Under TNFα stimulation, overexpression of the R59W variant resulted in increased phosphorylation of NF-κB p-65, IKK-α/β, IκBα (Ser32) compared to the wild type supporting the activation of the NF-ĸB signaling pathway, *p*-value < 0.05 (Figure 3A–C). In support of this, luciferase reporter assays also revealed significant activation (over threefold) of NF-κB p65 by overexpression of the R59W variant with TNFα stimulation (Figure 3D). Altogether, the presented data demonstrate that the R59W variant can potentiate TNFα mediated NF-κB signaling activation compared to the wild type.

### 3.7. Impact of NF-κB Activation on Other Inflammatory Cytokines

To assess and confirm the effect of NF-κB pathway activation, we measured the protein expressions of inflammatory cytokines, such as TNFα and IL6, regulated by the NF-κB pathway. There was a significant increase in the expression of both TNFα and IL6 (Figure 4A, * *p* < 0.05, n = 3)

### 3.8. Structural and Binding Analyses for the ANGPTL8 R59W Variant and the Wild Type

The data presented so far demonstrate that the ANGPTL8 R59W variant affects the cellular NF-κB activity. Therefore, understanding the structural changes in the mutated ANGPTL8 is critical to understanding its function and potential effects on cellular activity. ANGPTL8 structure modeled with I-TASSER and SWISS-Model demonstrates two α-helical subunits connected by a six-residue coil (Figure 4B). Furthermore, ANGPTL8 stability analysis using DynaMUT webserver showed that the R59W variant resulted in a more dynamic structure with a ΔΔG of −1.37 kcal/mol.

Zhang et al. have suggested that ANGPTL8 binding to IKKβ affects IKKβ cellular function [18]. Therefore, to examine whether the interaction of the ANGPTL8 R59W variant with IKKβ affected IKKβ downstream cellular function, we modeled both the ANGPTL8 wild type (R59) and mutant (W59) interaction with IKKβ (Figure 4C) to form an ANGPTL8-IKKβ complex using the HDOCK docking server [32] (Figure 4D,E). The binding affinities of ANGPTL8 R59 and W59 to IKKβ at 37 °C were calculated with the PRODIGY server. The binding affinity of ANGPTL8 R59 to IKKβ was 4.1 × 10^−8^ Kd/M, whereas ANGPTL8 W59 demonstrated a binding affinity of 3.7 × 10^−8^ Kd/M to IKKβ. Furthermore, the binding interface analysis of ANGPTL8-R59 and -W59 complexes with IKKβ showed different interaction interfaces, whereby ANGPTL8-R59 formed seven H-bonds with IKKβ compared to the six H-bonds formed with ANGPTL8-W59, which further corroborates the weaker binding affinity in the mutant complex with IKKβ in comparison to the wild type. Therefore, the weaker interaction demonstrated with the ANGPTL8-W59-IKKβ complex could positively impact NF-κB inflammatory activities presented downstream in comparison to the ANGPTL8-R59 complex with IKKβ (Figure 5).

### 3.9. Results from Power Calculation

Results from power calculation indicated that the discovery cohort of the study had 80% power to detect associations with the ANGPTL8 variant explaining 1% variance in the quantitative trait of TNFα in the discovery cohort and 2.8% variance in the replication cohort. The observed effect sizes of TNFα were 2.852 and 12.87 in the discovery and replication cohorts, respectively. These values are closer to the expected effect sizes of 3.271 and 12.628 as revealed by the power calculation for TNFα.

## 4. Discussion

ANGPTL8 plays an important role in lipid and glucose metabolism and is widely associated with various metabolic disorders, such as obesity and T2D, with more recent studies highlighting its role in inflammation [13,36]. However, the role of the *ANGPTL8* rs2278426 (p.R59W) variant remains unclear and needs to be explored in the context of inflammation. To our knowledge, we are the first to investigate the effect of the variant R59W on inflammation. Results from our genetic association tests and in vitro assays highlight the influential role of the ANGPTL8 R59W variant for regulating inflammation in hepatocytes. Finally, structural and binding analyses of the ANGPTL8 R59W variant revealed a less stable structure with weak transient binding compared to the wild type, which possibly explains the increased NF-κB activity in the in vitro analyses.

A similar R to W variant has been previously reported and was shown to disrupt functional domains within Troponin T or other proteins [37]. Our genetic association study has shown that there is an increase in the levels of proinflammatory makers, namely IL7, IL6, TNFα, and ghrelin, in the blood circulation of individuals having genotypes of alternate allele (CT+TT) as compared to reference homozygotes (CC) in a cohort of 867 individuals. Our previous cross-sectional study of 283 non-diabetic Arab individuals highlighted the significance of the R59W variant on glucose metabolism by way of showing that individuals having the R59W variant were associated with higher FBG levels compared to the wild type [38]. Previous studies have shown that circulating levels of ANGPTL8 were elevated in patients with severe infections and illustrated a strong correlation between ANGPTL8 and LPS-induced acute inflammatory response in animal models [12]. Circulating levels of ANGPTL8 were higher in human cohorts with NAFLD and it was further elucidated in the in vitro and in vivo models [17].

The role of ANGPTL8 in lipid metabolism has been well established. Studies show that that this genetic variant in *ANGPTL8* is associated with reductions in plasma levels of high-density lipoprotein-cholesterol (HDL-C) in European Americans, Hispanics, and African Americans, thus highlighting its role in lipoprotein/TG metabolism [37]. Likewise, other studies also confirmed the same R59W variant to contribute to lower HDL cholesterol levels in American Indians and Mexican Americans due to increased activation of ANGPTL3 by ANGPTL8 [39]. Another study of an Arab cohort suggested that the gene variants rs737337 (T/C) and rs2278426 (C/T) are associated with lower risk of hypercholesterolemia and hyperglycemia supporting the role of ANGPTL8 in lipid and glucose metabolism [40]. Additionally, it was shown in Japanese cohorts that the rates of T2D and impaired glucose tolerance were higher in subjects with the R59W variant; this observation highlights the role of the R59W variant in diabetes and thereby its use as a potential target for prevention of T2D [41]. No other group has ever studied the impact of this variant on inflammation.

Emerging evidence suggests that ANGPTL8 may play a role in inflammatory processes. Several studies have investigated the association between ANGPTL8 and inflammation. It is important to consider the complexity and limitations of the studies that pointed to the potential involvement of ANGPTL8 in inflammation. Further research is required to elucidate the exact mechanisms and the extent of contribution from ANGPTL8 to inflammation in various physiological and pathological contexts. Zhang et al. [18] have shown that ANGPTL8 can work intracellularly as a negative feedback inhibitor of NF-κB activation via self-associating its N-terminal region and interacting with Sequestosome-1 (p62/SQSTM1). The resulting ANGPTL8-p62 complex aggregates and acts as a platform in the recruitment and selective autophagic degradation of IκB kinase gamma (IKKγ/NEMO) [42]. On the other hand, the results of Liao et al. [23] indicated that ANGPTL8 plays an adverse role in the progression of intervertebral disc degeneration (IDD) and that the silencing of ANGPTL8 could reduce ECM degradation and inflammation in human NP cells via inhibition of the NF-κB signaling pathway. These differences were thought to be due to the differences in the used cell lines and the acute nature of stimulation in the first study versus the second. ANGPTL8 could possibly have pleiotropic functions in different tissues and organs; while it acts as a defense mechanism in certain instances, it is rather proinflammatory in chronic inflammation. Our results support the pro-inflammatory role of ANGPTL8 and its variant through modulating the NF-κB signaling pathway. Further work is warranted to fully explore the inflammatory role of ANGPTL8 and its variant under different stimulations and in various tissues. Our study further explored the inflammatory pathways linked with the overexpression of the wild type and variant (R59W) in hepatocytes with/without TNFα stimulation. Activity levels of NF-κB p65 were higher in the R59W variant as compared to the ANGPTL8 wild type and it was further elevated with TNFα stimulations for this variant (Figure 2B,C and Figure 3A–C). This data was further supported by the luciferase activity (Figure 2A and Figure 3D), highlighting the impact of ANGPTL8 R59W over the wild type in increasing the activity levels of NF-κBp65 which could further impact the inflammatory signaling cascade. In the presence of the overexpressed R59W variant and under the influence of inflammatory stimuli like TNFα, the activity levels of IKKα/β were enhanced which sequentially phosphorylates the NF-κB inhibitor IκBα and triggers its rapid degradation through the β-TrCP–derived ubiquitin-proteasome pathway resulting in the release of the NF-κB heterodimer from IκBα. No longer repressed by IκBα, the NF-κB heterodimer translocates to the nucleus from the cytoplasm, binds to its downstream target DNA sequence, and induces the expression of many markers involved in immune response as represented in our hypothetical model (Figure 5). This throws light on the role of R59W-ANGPTL8 in modulating inflammation through the NF-κB signaling pathway due to the exposure of inflammatory stimuli.

Next, we have shown that IL6 and TNFα expression was consistently upregulated when the R59W was overexpressed and stimulated by TNFα at the translational level depicting the proinflammatory role of this variant in hepatocytes (Figure 4A). This observation is corroborated by previous studies which have shown that IL6 can be induced in hepatocytes by LPS treatment through increased translocation of the P65 subunit of NF-κB to the nucleus at 1 hr of treatment [43]. A similar study showed that the influence of IL17 with the synergetic role of TNFα contributes to the development of inflammatory liver diseases in primary hepatocytes [44]. Another study indicated that ANGPTL8 promoted the TNFα-induced ECM degradation and inflammation through the hyperactivation of the NF-κB signaling pathway in nucleus pulposus (NP) cells [23]. Based on this evidence, we speculate that the R59W overexpression and stimulation with proinflammatory cytokines like TNFα could potentiate p-IKKα/β activity, inhibiting IκBα and subsequently promoting enhanced nuclear translocation of NF-κB p65 in the R59W variant and its downstream targets.

Previous studies have suggested that ANGPTL8 interaction with IKKα/β is necessary for IKKα/β downstream cellular activity. Furthermore, in this study, we demonstrated that the ANGPTL8 W59 variant enhanced NF-κB activity compared to the wild type under TNFα stimulation as a result of IKKα/β phosphorylation. Structural modeling of the ANGPTL8 wild type and the R59W variant presented a helical structure wherein the helices are joined by a loop as previously shown by Siddiqa et al. [29]. Stability analysis of the ANGPTL8 R59W variant presented a less stable structure than the wild type, whereby such substitutions have been shown to affect protein-protein interaction and lead to differential downstream activities [45]. Moreover, the effect of the ANGPTL8 W59 variant on the cellular pathway is corroborated by our interaction analysis, which showed that the wild type has a higher binding affinity of 4.1 × 10^−8^ Kd/M to IKKβ compared to ANGPTL8 W59 (3.7 × 10^−8^ Kd/M). This difference in ANGPTL8 WT and W59 binding affinities to IKKβ can influence its cellular activity as it has been previously suggested that ANGPTL8 acts as a transient protein for IKKβ activity [18].

At the molecular level, protein-protein interactions are critical in almost every function in the cell. As such, protein-protein interactions are imperative in understanding cell function and diseases. Protein-protein interactions are classified as two categories, transient and permanent, whereby permanent interactions are two protein partners that form a strong complex and continue to function without breaking apart. On the other hand, with transient interaction, the protein-protein interaction forms a weak complex for a short period and then dissociates [46,47,48]. Proteins interacting in a weakly transient manner show a fast bound-unbound equilibrium as observed with the ANGPTL8-W59-IKKβ complex, resulting in more differential and possibly pronounced downstream activities than the ANGPTL8 -WT- IKKβ complex.

To conclude, our studies demonstrate the intracellular role of the ANGPTL8 R59W variant in regulating proinflammatory events as reflected by genetic association tests, in vitro assays, and structural studies. Further studies are warranted to unravel the other signaling pathways and the crosstalk between the ligand-receptor interaction to deepen our knowledge of the mechanistic role of the variant R59W in the context of inflammation. The findings of this study could provide insight into the role of the ANGPTL8 R59W variant in influencing inflammation events/cascade. Thus, by modulating the NF-κB signaling pathway through the ANGPTL8 R59W variant, it would be possible to improve immune response and cell fate and thereby to modulate inflammatory diseases.

## Figures and Tables

**Figure 1 cells-12-02563-f001:**
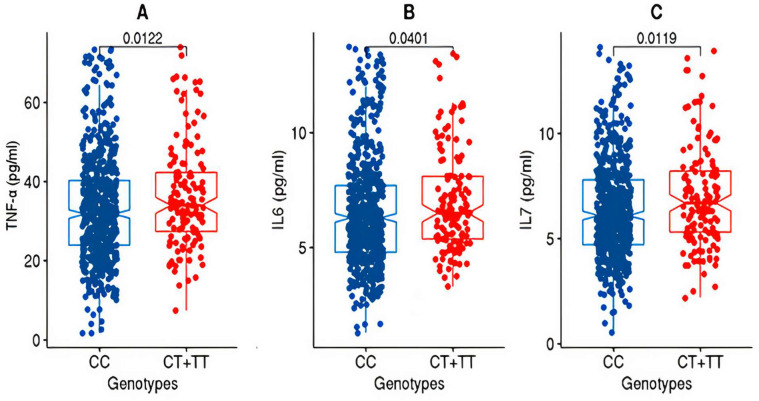
Box plots for TNFα, IL6, and IL7 from the discovery cohort. The figure shows the distribution of TNF-alpha (**A**), IL6 (**B**), and IL7 (**C**) levels in two genotype groups: “CC” and “CT+TT” in the discovery cohort. The “CC” group exhibits lower levels of TNF-alpha, IL6, and IL7 compared to the “CT+TT” group. The statistical significance of the difference between the groups was assessed using the Wilcoxon rank-sum test (Mann–Whitney U test), and the resulting *p*-value was <0.05.

**Figure 2 cells-12-02563-f002:**
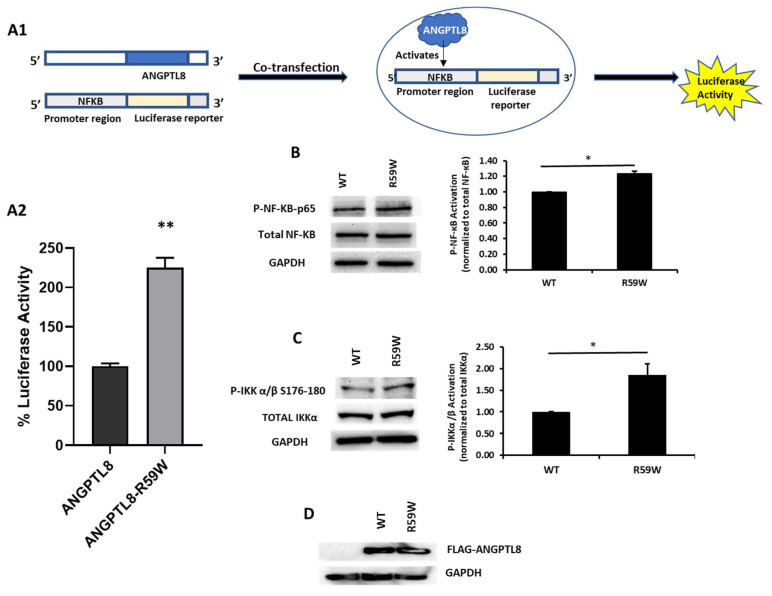
The R59W variant activates the NF-κB pathway. (**A**) Cartoon model for NF-κb reporter plasmid (**A1**) and luciferase assay illustrating overexpression of R59W variant showing increased activation of NF-κB p65 as compared to the wild type (**A2**). (**B**) illustrates increased phosphorylation of P- NF-κB-P65. (**C**) illustrates increased phosphorylation of P-IKK α/β S176-180. (**D**) represents image of PCMV6, ANGPTL8 flag, and its variant after transfection by Western blot. The data represent the Mean +/− SEM of three independent experiments, * *p* < 0.05, ** *p* < 0.001, as determined using Student’s *t*-test, n = 3 (original gel images are available in the Appendix A).

**Figure 3 cells-12-02563-f003:**
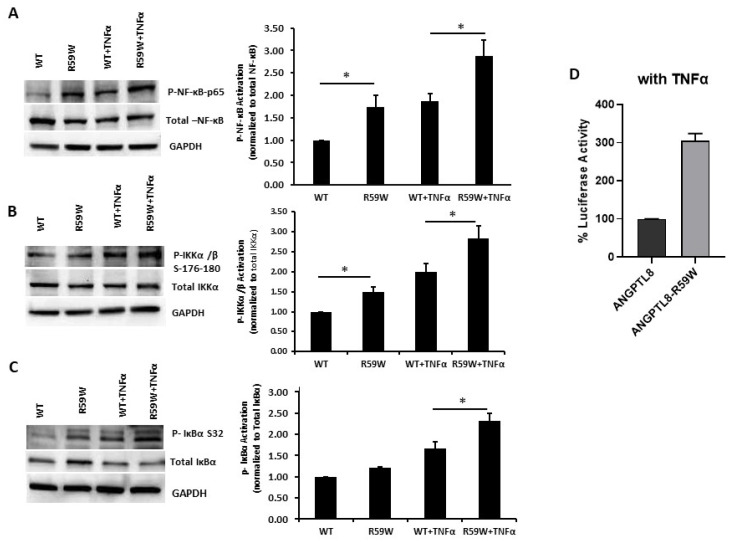
R59W variant activates the NF-κB pathway under TNFα stimulation compared to the wild type. (**A**–**C**) illustrate overexpression and TNFα stimulation of R59W further increasing the phosphorylation levels of NF-κB-P65, IKK α/β S176-180, IκBα (Ser32). (**D**) illustrates further activation of NF-κB p65 in the luciferase reporter assays when stimulated by TNFα compared to the wild type. The data represent the Mean +/− SEM of three independent experiments, * *p* < 0.05 as determined using student’s *t*-test, n = 3.

**Figure 4 cells-12-02563-f004:**
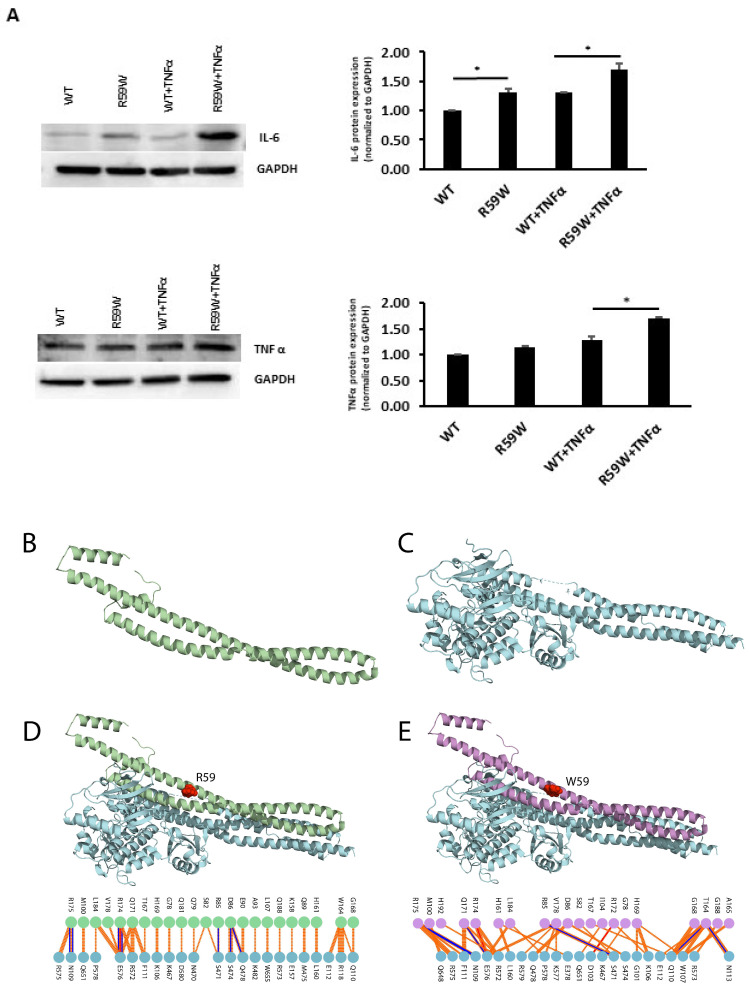
(**A**). Impact of overexpression of the R59W variant with/without TNFα stimulation on IL6 and TNFα, as represented by western blots. The data represent the Mean +/− SEM of three independent experiments, * *p* < 0.05 as determined using Student’s *t*-test, n = 3. (**B**–**E**). Structural Analysis of ANGPTL8 and IKKβ. (**B**) depicts the modeled structure of ANGPTL8 with I-TASSER and Swiss-Model; (**C**) depicts the crystal structure of IKK monomer (PDB ID: 4KIK); D depicts the modeled ANGPTL8 wild type (green) with IKKβ; and (**E**) depicts the modeled ANGPTL8 W59 (purple) with IKKβ.

**Figure 5 cells-12-02563-f005:**
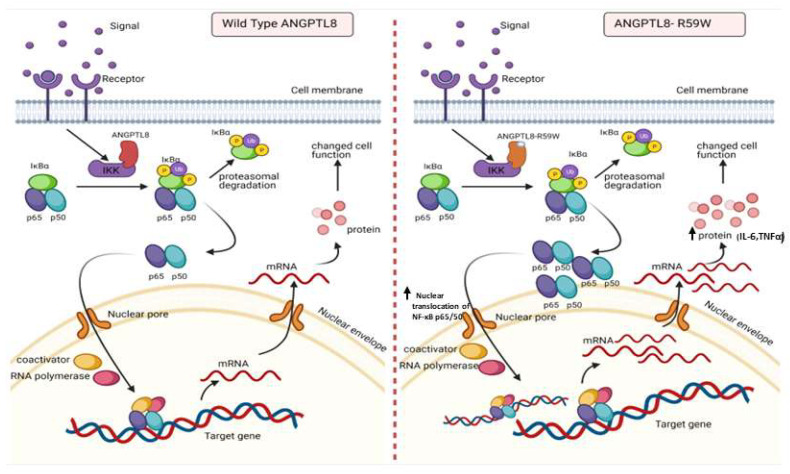
A proposed model depicting the impact of overexpressing the ANGPTL8 R59W variant in influencing the NF-κB pathway. After the overexpression of the wild type ANGPLT8/ANGPTL8-R59W and under the influence of inflammatory stimuli like TNFα, the activity levels of IKKα/β were enhanced which sequentially phosphorylates the NF-κB inhibitor, IκBα, and triggers its rapid degradation through the proteasomal degradation. This releases the NF-κB heterodimer as it is no longer repressed by IκBα. Subsequently, the NF-κB heterodimer translocates from cytoplasm to nucleus, binds to its downstream target DNA sequence, and induces the expression of proinflammatory proteins. This expression of proinflammatory cytokines is higher in the overexpressed ANGPTL8 R59W variant as compared to the wild type ANGPTL8 under stimulation due to its increased nuclear translocation of NF-κB heterodimers. This model depicts the proinflammatory role of R59W-ANGPTL8 in modulating the inflammation through the NF-κB signaling pathway.

**Table 1 cells-12-02563-t001:** (**A**) Clinical characteristics of the study cohorts. Values are presented as mean ± SD. (**B**) Results of statistical tests for associations between the *ANGPTL8* rs2278426 (R59W) variant and biomarkers. Only those associations that showed significant *p*-values (≤0.05) in the discovery cohort are shown.

(**A**)					
**Phenotypes**		**Discovery Cohort**		**Replication Cohort**	
Male:Female		516:351 (59.5%:40.5%)		125:153 (45%:55%)	
Age (years)		43.36 ± 10.78		46.25 ± 12.38	
Height (meters)		1.64 ± 0.86		1.64 ± 0.09	
Weight (kilograms)		77.95 ± 15.02		81.40 ± 16.23	
Body mass index, BMI (kg/m^2^)		28.53 ± 4.9		29.93 ± 5.17	
Waist circumference, WC (cm)		93.78 ± 11.51		99.36 ± 13.36	
High-density lipoprotein, HDL (mmol/L)		1.15 ± 0.29		1.20 ± 0.32	
Total cholesterol, TC (mmol/L)		5.19 ± 0.93		5.02 ± 1.09	
Low-density lipoprotein, LDL (mmol/L)		3.34 ± 0.89		3.13 ± 0.96	
Triglyceride, TG (mmol/L)		1.42 ± 0.61		1.22 ± 0.59	
Fasting plasma glucose, FPG (mmol/L)		5.14 ± 0.75		5.77 ± 1.24	
Haemoglobin A1c, HbA1c (%)		5.47 ± 0.74		6.31 ± 1.29	
Obese versus non-obese		331:536		136:142	
Diabetics versus non-diabetics		259:608		121:157	
Hypertensive versus non-hypertensive		364:427		84:192	
Diabetes medication (yes versus no)		197:670		101:175	
Lipid lowering medication (yes versus no)		197:670		89:189	
(**B**)					
**Traits**	**Cohorts**	**Correction ^@^**	**Sample Size**	**Beta**	** *p* ** **-Value**
Tumor necrosis factor alpha, TNFα	Discovery	R	738	2.852	0.0147
		R+OS	738	1.702	0.0510
		R+DS	738	2.852	0.0148
		R+HS	670	3.501	0.0028
	Replication	R	166	12.87	0.0228
		R+OS	166	12.98	0.0208
		R+DS	166	12.5	0.0273
		R+HS	166	11.58	0.0397
Interleukin 7, IL7	Discovery	R	799	2.284	0.03705
		R+OS	799	2.286	0.037
		R+DS	799	2.243	0.04052
		R+HS	660	0.6374	0.00224
	Replication	R	162	2.156	0.03333
		R+OS	162	2.156	0.0338
		R+DS	162	2.062	0.0394
		R+HS	162	1.897	0.0566
Interleukin 6, IL6	Discovery	R	735	0.4592	0.02612
		R+OS	735	0.4619	0.02489
		R+DS	735	0.4592	0.02622
		R+HS	670	0.494	0.02086
	Replication	R	166	1.334	0.1267
		R+OS	166	1.328	0.1294
		R+DS	166	−1.616	0.1554
		R+HS	166	1.179	0.1743
Ghrelin	Discovery	R	768	139.7	0.00327
		R+OS	768	135	0.00405
		R+DS	768	140.3	0.00316
		R+HS	696	141.6	0.00475
	Replication	R	161	37.43	0.3155
		R+OS	161	38.89	0.2973
		R+DS	161	38.5	0.305
		R+HS	161	37.7	0.3179

^@^, R indicates regular correction for age and sex; R+OS indicates correction for age, sex, and obesity status; R+DS indicates correction for age, sex, and diabetes status; R+HS indicates correction for age, sex, and hypertension status.

**Table 2 cells-12-02563-t002:** SNP quality assessment tests on the ANGPTL8 rs2278426 (R59W) for HWE.

Cohort	Minor/Major Alleles	MAF ^@^	Genotype Counts	Observed Heterozygous	Expected Heterozygous	*p*-Value HWE ^$^
Discovery	T/C	0.102	11/155/701	0.178	0.183	0.457
Replication	T/C	0.101	5/46/227	0.166	0.181	0.173

^@^ MAF, minor allele frequency, ^$^ HWE, Hardy-Weinberg Equilibrium.

**Table 3 cells-12-02563-t003:** Logistic regression analysis for the impact of the variant rs2278426_T on the disease status of the participants in the cohorts.

Disease Status	Cohorts	OR [95% CI] ^@^	Standard Error	*p*-Value
Diabetes status	Discovery cohort	1.15 [0.80–1.64]	0.181	0.442
	Replication cohort	1.22 [0.68–2.19]	0.296	0.486
Obese status	Discovery cohort	0.898 [0.64–1.24]	0.166	0.521
	Replication cohort	0.93 [0.54–1.61]	0.274	0.812
Hypertension status	Discovery cohort	1.06 [0.76–1.49]	0.171	0.709
	Replication cohort	1.27 [0.68–2.35]	0.315	0.447

^@^ OR, odds ratio, and 95% confidence interval.

## Data Availability

All data relevant to the study are included in the article. Further data are available upon reasonable request.

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
