# Peer review of "The Proinflammatory Role of ANGPTL8 R59W Variant in Modulating Inflammation through NF-κB Signaling Pathway under TNFα Stimulation"

_cells, 2023, doi:10.3390/cells12212563_

Round 1

Reviewer 1 Report

Comments and Suggestions for Authors

The authors investigated the impact of an ANGPTL8 single nucleotide polymorphism on inflammatory biomarkers and NFkB activity. Therefore, they genotyped Arab individuals and associated the genotype with inflammatory markers. Activity of several members of the NFkB pathway was measured in a hepatocyte cell line.

The results showed that an ANGPLT8 variant is associated with increased levels of the circulating inflammatory cytokines TNFa and IL7. Proteins in the classical NFkB activation pathway also showed increased activity. These effects could at least partially be related to a weaker binding capacity of ANGPTL8 to IKKa.

The study is interesting and shows new results concerning effects of the ANGPTL variant on inflammatory pathways. However, there are several points that need to be respected.

Specific points:

·         Abstract: There are a lot of abbreviations in the abstract which should be avoided.

·         In general, there are several non-common abbreviations which are not properly explained (e.g. WC, FPG, HWE).

·         In the abstract, it does not become clear to what protein ANGPTL8 binds in the NFkB cascade.

·         Introduction: It is not indicated why the authors focussed on the R59W variant of ANGPTL8. Is this a common variant and is it particularly occurring in Arab individuals?

·          The authors indicate an allele frequency of the R59W variant of 0.0102 in the discovery cohort and 0.101 in the replication cohort. What is the reason for the 10fold difference?

·         Although the allele frequency is indicated, it would be helpful to provide the absolute numbers of individuals with wild type and mutated alleles in the text.

·         The cohorts differ strongly in their HbA1c ratios. From these values, individuals in the discovery cohort showed normal blood glucose levels while individuals in the replication cohort show at least disturbed glucose tolerance or pre-diabetes. In my opinion, this is problematic. The difference is also apparent for obesity and medication. The authors tried to perform a correction, however, in table 1b the samples sizes are always the same irrespective of the different groups. This should be clarified.

·         Figures 2-4: How did the authors manage to detect and quantify phosphorylated and total proteins on the same Blot? All original Blots which have been used for analysis should be provided.

·         How did the authors calculate the rate of phosphorylation? In the diagrams, they indicate that they normalized against GAPDH, but did they also normalize p-protein/total protein?

·         It should be indicated how data are presented eg. Mean +/- SEM. For the diagrams, it would be preferable to show the individual values in the columns.

·         Figure 3D is cut.

·         It is not really clear why a weaker interaction of ANGPTL8 with IKKa should increase NFkB activity.

·         Some symbols are not correctly displayed.

Comments on the Quality of English Language

There are some writing mistakes, however, in general the manuscript is well written.

Author Response

The authors investigated the impact of an ANGPTL8 single nucleotide polymorphism on inflammatory biomarkers and NFkB activity. Therefore, they genotyped Arab individuals and associated the genotype with inflammatory markers. Activity of several members of the NFkB pathway was measured in a hepatocyte cell line.

The results showed that an ANGPLT8 variant is associated with increased levels of the circulating inflammatory cytokines TNFa and IL7. Proteins in the classical NFkB activation pathway also showed increased activity. These effects could at least partially be related to a weaker binding capacity of ANGPTL8 to IKKa.

The study is interesting and shows new results concerning effects of the ANGPTL variant on inflammatory pathways. However, there are several points that need to be respected.

Specific points:

  1. Abstract: There are a lot of abbreviations in the abstract which should be avoided.

We thank the reviewer for the positive comments. As pointed out by the reviewer, we have removed some of the unnecessary abbreviations and detailed the rest.

  1. In general, there are several non-common abbreviations which are not properly explained (e.g. WC, FPG, HWE).

We have now expanded the non-common abbreviations throughout the manuscript and in Tables.

  1. In the abstract, it does not become clear to what protein ANGPTL8 binds in the NFkB cascade.

Zhang et al., have shown ANGPTL8 interacting with IKK β and É£ through various Co-immunoprecipitation (co-IP) experiments. Therefore, we have edited the sentence in the abstract and it reads as follow:

“It interacts with ANGPTL3 and ANGPTL4 to regulate LPL activity, and with IKK to modulate NF-κB activity.”

Reference:

Zhang, Y., Guo, X., Yan, W. et al. ANGPTL8 negatively regulates NF-κB activation by facilitating selective autophagic degradation of IKKγ. Nat Commun 8, 2164 (2017). https://doi.org/10.1038/s41467-017-02355-w

  1. Introduction: It is not indicated why the authors focussed on the R59W variant of ANGPTL8. Is this a common variant and is it particularly occurring in Arab individuals?

We thank the reviewer for this useful point and now added the following text to the manuscript at page 3.

“Several genome-wide association studies, as listed in the GWAS Catalog (https://www.ebi.ac.uk/gwas/), have associated the ANGPTL8 rs2278426_ c.175C.T_p.R59W variant with the lipid traits of total cholesterol, low-density cholesterol, high-density cholesterol, and triglycerides in global populations such as those of European ancestry, East Asian ancestry, Hispanic ancestry as well as in multi-ethnic cohorts which include various additional ethnicities such as African American and south Asian. The variant is also listed in GWAS Catalog as associated with waist-hip ratio in individuals of European ancestry. Although the role of ANGPTL8 was established in the context of inflammation, the impact of R59W ANGPTL8 variant in regulating inflammation in vitro has not been explored so far to our knowledge. Therefore, our objective was to investigate the differential effect of ANGPTL8 R59W variant over the wild type (WT) in modulating inflammatory pathways.”

  1. The authors indicate an allele frequency of the R59W variant of 0.0102 in the discovery cohort and 0.101 in the replication cohort. What is the reason for the 10 fold difference?

Thank you for pointing out this typo. It should be 0.102. It was correctly mentioned in Table 2. Now we corrected it in the main text in page 6. The corrected text reads as follow:

“In our study cohorts, the minor allele (T) occurred at a frequency of around 0.102 in the discovery cohort and 0.101 in the replication cohort”.

  1. Although the allele frequency is indicated, it would be helpful to provide the absolute numbers of individuals with wild type and mutated alleles in the text.

We have now added the following text to page 6 in the manuscript:

 “Distribution of individuals in terms of genotypes homozygous for reference allele (CC): heterozygous for reference and mutated allele (CT): homozygous for mutated allele (TT) was 701:155:11 in discovery cohort and 227:46:5 in replication cohort”.

  1. The cohorts differ strongly in their HbA1c ratios. From these values, individuals in the discovery cohort showed normal blood glucose levels while individuals in the replication cohort show at least disturbed glucose tolerance or pre-diabetes. In my opinion, this is problematic. The difference is also apparent for obesity and medication. The authors tried to perform a correction, however, in table 1b the samples sizes are always the same irrespective of the different groups. This should be clarified.

The two cohorts (discovery and replication) are as mentioned in column 2. The sizes of these two cohorts are different as you can see in the column of sample size. The sample sizes of a cohort (e.g. discovery cohort) for the four models implemented for different corrections (R, regular correction for age and sex; R+OS for regular and additional correction for obesity status; R+DS for regular and additional correction for diabetes status; and R+HS for regular and additional correction for hypertension) will tend to be same; the only exception seen is for the R+HS correction as the hypertension status was not available for some of the participants.

  1. Figures 2-4: How did the authors manage to detect and quantify phosphorylated and total proteins on the same Blot? All original Blots which have been used for analysis should be provided.

We appreciate the reviewer’s concern. Below is a clarification for this point:

“After visualizing the phosphorylated protein on the fresh blot, the same blot is stripped using the stripping buffer (Millipore, Catalog Number 2504), followed manufacturers’ recommendation. It is then re-probed to visualize the total protein of the same using the different primary antibody/secondary antibody. Finally, the same blot is stripped again, re-probed for housekeeping protein (GAPDH)using the same methodology. A blot is used maximum up to 3 times and re-probed for different antibodies. This saves much time without the requirement for multiple gels and transfers and the same blots can be used for multiple target proteins.” All original Blots were uploaded during the submission.

  1. How did the authors calculate the rate of phosphorylation? In the diagrams, they indicate that they normalized against GAPDH, but did they also normalize p-protein/total protein?

We have normalized the p-proteins to their corresponding total proteins as well as GAPDH. We have corrected the diagrams to reflect on the normalization to total proteins. The band intensities were measured using Quantity one software (Bio-Rad, USA) using the Versadoc imaging system (Bio-Rad, USA).

  1. It should be indicated how data are presented eg. Mean +/- SEM.

We now added legend to Table 1A indicating that the values are presented as mean±SD.

For the Western blot analyses, the data represent the Mean+/-SEM of three independent runs, *P<0.05, **P<0.001 and these details have now been added to the revised version of the manuscript.

  1. Figure 3D is cut.

It was accidentally cut during the conversion from Word to PDF. It is currently corrected.

  1. It is not really clear why a weaker interaction of ANGPTL8 with IKKa should increase NFkB activity.

As a clarification for this point, we have added the following paragraph to the discussion in lines 555-564 of page 16:

“At the molecular level, protein-protein interactions are critical in almost every function in the cell. As such, protein-protein interactions are imperative in understanding cell function and diseases. Protein-protein interactions are divided into two categories, transient and permanent, whereby permanent interactions are two protein partners that form a strong complex and continue to function without breaking apart. On the other hand, with transient interaction, the protein-protein interaction forms a weak complex for a short period and then dissociates [46-48]. Proteins interacting in a weakly transient manner show a fast bound-unbound equilibrium as observed with ANGPTL8-W59-IKKβ complex, resulting in more differential and possibly pronounced downstream activities than ANGPTL8 -WT- IKKβ complex.”

  1. Some symbols are not correctly displayed.

    We thank the reviewer for pointing this out and all wrongly displayed symbols were corrected in the revised manuscript.

Reviewer 2 Report

Comments and Suggestions for Authors

Mohamed Abu-Farha et al. demonstrated the R59W mutation of ANGPTL8 is associated with elevated inflammation and validated the inflammation is through affecting the NF-KB pathway. The study was appropriately designed and experiments were properly conducted. However, I still have some comments.

The title could be more conclusive than the ambiguously described in the present form.

In section 2.9, more information about the design of NF-kB reporter plasmid is needed. You can add it to the supplement if needed.

In Figure 1, what statistical test was used and more detailed information is needed.

In Figure 2, move the cartoon to the beginning of the figure. Based on the cartoon, it seems that ANGPTL8 is interacting with NF-KB elements directly, which needs clarification or modification if not. Where is the empty vector which is also critical to show the effect of expression of wild type? Did authors examine the translocation of P65 from cytoplasm to nucleus which is the process for the activation of NF-kB pathway? In Figure 5, the authors also showed the more P65 release from inhibitor.

In Figure 4, ANGPTL8 facilitates the ANGPTL8-p62-IKKγ complex formation, leading to IKKγ degradation and NF-κB activation. Here, the authors claimed the weak binding affinity promotes the NF-kB activation by enhancing the degradation of the inhibitor of NF-kB. Is there any basis for this assumption or claim?

More detailed figure legend is needed in Figure 5. It’s better to highlight the differentiated parts between two pathways.

Three-line table is preferred.

Full names are missing in Table 1. Lines 316-319 are duplicates of lines 303-307.

It’s advised to add the conclusion that the population meets the assumption of HW equilibrium in addition to the test result in line 308

Author Response

Mohamed Abu-Farha et al. demonstrated the R59W mutation of ANGPTL8 is associated with elevated inflammation and validated the inflammation is through affecting the NF-KB pathway. The study was appropriately designed and experiments were properly conducted. However, I still have some comments.

  1. The title could be more conclusive than the ambiguously described in the present form.

We thank the reviewer for the positive comment on the study. As per reviewer’s recommendation, we have now changed the title to the following:

“The proinflammatory role of ANGPTL8 R59W variant in modulating inflammation through NF-κB Signalling pathway under TNFα stimulation”

  1. In section 2.9, more information about the design of NF-kB reporter plasmid is needed. You can add it to the supplement if needed.

As suggested by the reviewer, we have added sentences on the design of NF-kB reporter plasmid and provided the reference.

Briefly, plasmids 3xwt-κB-pGL3 and 3xmut-κB-pGL3 were constructed by inserting three copies of wild type (5’-AGTTGAGGGGACTTTCCCAGGCTG-3’) NF-κB binding site into the unique Nhe-I and Xho-I restriction sites upstream of the SV40 minimal promoter of pGL3 vector (Promega, Madison, WI).

References:

al-Haj, L., Al-Ahmadi, W., Al-Saif, M. et al. Cloning-free regulated monitoring of reporter and gene expression. BMC Molecular Biol 10, 20 (2009). https://doi.org/10.1186/1471-2199-10-20

  1. In Figure 1, what statistical test was used and more detailed information is needed.

We have now added the following legends to Figure 1. “The figure shows the distribution of TNF-alpha, IL-6 and IL-7 levels in two genotype groups: "CC" and "CT+TT" in the discovery cohort. The "CC" group exhibits lower levels of TNF-alpha, IL-6 and IL-7 compared to the "CT+TT" group. The statistical significance of the difference between the groups was assessed using the Wilcoxon rank-sum test (Mann-Whitney U test), and the resulting p-value was <0.05”.

  1. In Figure 2, move the cartoon to the beginning of the figure. Based on the cartoon, it seems that ANGPTL8 is interacting with NF-KB elements directly, which needs clarification or modification if not. Where is the empty vector which is also critical to show the effect of expression of wild type? Did authors examine the translocation of P65 from cytoplasm to nucleus which is the process for the activation of NF-kB pathway? In Figure 5, the authors also showed the more P65 release from inhibitor.

       As suggested, we have moved the cartoon to the beginning of the Figure 2. As a clarification and based on the luciferase assay finding, it is possible that ANGPTL8 can activate NF-kB activity through direct or indirect binding to NF-kB binding sites.

We have studied the effect of the empty vector over wildtype and its variant during the initial phase of the study as shown below. However, our hypothesis was based on learning the differential effect of R59W Angplt8 variant as compared to the wild type Angplt8 both in vivo and in vitro. Hence, the reference/benchmark to investigate the inflammatory role of Angptl8 R59W would be the wild type Angptl8, although the effect of  empty vector(PCMV6) on Wildtype Angptl8/R59W Angptl8 has been studied as shown below:

We appreciate the reviewer comment regarding the nuclear translocation assessment. However, it was not done. Further, phosphorylation of IκBα (NF-κB inhibitor) is known to lead for its sequential proteasomal degradation and the release of the repressed P65 monomer.

  1. In Figure 4, ANGPTL8 facilitates the ANGPTL8-p62-IKKγ complex formation, leading to IKKγ degradation and NF-κB activation. Here, the authors claimed the weak binding affinity promotes the NF-kB activation by enhancing the degradation of the inhibitor of NF-kB. Is there any basis for this assumption or claim?

As a clarification for this point, we have added the following paragraph to the discussion in lines 555-564 of page 16:

“At the molecular level, protein-protein interactions are critical in almost every function in the cell. As such, protein-protein interactions are imperative in understanding cell function and diseases. Protein-protein interactions are divided into two categories, transient and permanent, whereby permanent interactions are two protein partners that form a strong complex and continue to function without breaking apart. On the other hand, with transient interaction, the protein-protein interaction forms a weak complex for a short period and then dissociates [46-48]. Proteins interacting in a weakly transient manner show a fast bound-unbound equilibrium as observed with ANGPTL8-W59-IKKβ complex, resulting in more differential and possibly pronounced downstream activities than ANGPTL8 -WT- IKKβ complex.”

Further, phosphorylation of IκBα (NF-κB inhibitor) is known to lead for its sequential proteasomal degradation and the release of the repressed NF-kB-P50/P65 heterodimer. More IκBα phosphorylation was observed in the ANGPTL8-W59 cell compared to wild type under TNFα stimulation.

  1. More detailed figure legend is needed in Figure 5. It’s better to highlight the differentiated parts between two pathways. Three-line table is preferred.

We have now added a more detailed explanation to figure 5 legend and highlighted the differentiated parts in the modified figure 5 in page 13:

After the overexpression of Wild type ANGPLT8/ ANGPTL8-R59W and under the influence of inflammatory stimuli like TNFα, the activity levels of IKKα/β were enhanced which sequentially phosphorylate the NF-κB inhibitor, IκBα and trigger its rapid degradation through the proteasomal degradation. This releases the NF-κB heterodimer as it is no longer repressed by IκBα. Subsequently, the NF-κB heterodimer translocates from the cytoplasm to the nucleus, binds to its downstream target DNA sequence, and induces the expression of proinflammatory proteins. This expression of proinflammatory cytokines is higher in the overexpressed Angptl8 R59W variant as compared to the wild type ANGPTL8 under stimulation due to its increased nuclear translocation of NF-κB heterodimers. This model depicts the proinflammatory role of R59W-Angplt8 in modulating the inflammation through NF-κB signaling pathway.”

  1. Full names are missing in Table 1. Lines 316-319 are duplicates of lines 303-307.

We have now provided the full names in Table 1a and 1b.

We now removed the duplicate text from the table legends.

  1. It’s advised to add the conclusion that the population meets the assumption of HW equilibrium in addition to the test result in line 308

We have now added the phrase “and thus the population meets the assumption of HW equilibrium” to line 335 in the manuscript.

Round 2

Reviewer 1 Report

Comments and Suggestions for Authors

Please, see attachment.

Comments on the Quality of English Language

There are some minor writing mistakes which should be corrected.

Author Response

We would like to thank the reviewer for the comments and feedback, and we hope that our responses and explanation will be satisfactory.

Our reblot stripping solution (Cat # 2504, Millipore) is a strong stripping solution that effectively removes the antibodies from the blot after development/exposure. Below are some of the blots that were stripped using their 1x reconstituted buffer at high speed on a shaker for 10-15 minutes for the first time of stripping and for 20-25 minutes for the second time of stripping. Our primary antibody used for detecting the target protein of interest and GAPDH is very specific to the band of interest. This is depicted in the band density, shape and size of the band that shows up in the same blot. Some carryovers may be seen in the blots due to similarity in the host species of primary antibody.  Nonetheless, the band of interest shows up with greater intensity and is represented in the images below (attached).

The carryover bands and cross reactivity from the phospho to the total protein and from the total protein to the GAPDH is very minimal.

Moreover, and as requested by the reviewer, all the replica raw files used in the analyses have been provided. Further and suggested by the reviewer, we had our revised manuscript read and corrected by native speaker.

Reviewer 2 Report

Comments and Suggestions for Authors

Thanks for the responses.

Most comments were addressed.

I would suggest having the original gel images without cropping included in the supplementary. This may help remove any concerns. 

Overall the supplemental figures look good except that figures 2B, 3A, 3B, and 3C look different from the previous/present draft, which may be emphasized. 

Besides that, the number of tests (n=3) was not indicated in the supplement. 

Author Response

We thank the reviewer for the constrictive comments that will definitely improve our manuscript.

We do not mind adding the original images as a supplementary file.
In fact, we have added the following sentence in the attached revised manuscript (line 373):
"original gel images are available in the supplementary".

Round 3

Reviewer 1 Report

Comments and Suggestions for Authors

thank you for your mail and the files. In the supplementary files the authors added complete Western Blot images of the Blots shown in the manuscript. This Blots looked fine. However, they still show only one of three experiments and, since SEMs are relatively high it would be really of interest to see all of the Blots. The fact that the authors used SD for all experiments except the Western Blots is somehow strange. Sorry for still insisting on this point but the authors still did not deliver what has been asked for.

Comments on the Quality of English Language

There are some minor writing mistakes which should be corrected.